# Barriers and facilitators to neonatal health and care-seeking behaviours in rural Cambodia: a qualitative study

Gabriella Watson [ORCID],[1] Kaajal Patel,[1,2] Daly Leng,[1] Dary Vanna,[1] Sophanou Khut,[1] Manila Prak,[1] Claudia Turner[2,3,4]

[1]Global Child Health Department, Angkor Hospital for Children, Siem Reap, Cambodia
[2]Mahidol Oxford Tropical Medicine Research Unit, Mahidol University, Bangkok, Thailand
[3]University of Oxford Centre for Tropical Medicine and Global Health, University of Oxford, Oxford, UK
[4]Cambodia Oxford Medical Research Unit, Angkor Hospital for Children, Siem Reap, Cambodia

**Correspondence to**
Dr Gabriella Watson;
gabriellawatson@hotmail.com

## ABSTRACT

**Objectives** Neonatal mortality remains persistently high in low-income and middle-income countries. In Cambodia, there is a paucity of data on the perception of neonatal health and care-seeking behaviours at the community level. This study aimed to identify influencers of neonatal health and healthcare-seeking behaviour in a rural Cambodian province.

**Design** A qualitative study using focus group discussions and thematic content analysis.

**Setting** Four health centres in a rural province of Northern Cambodia.

**Participants** Twenty-four focus group discussions were conducted with 85 community health workers in 2019.

**Results** Community health workers recognised an improvement in neonatal health over time. Key influencers to neonatal health were identified as knowledge, sociocultural behaviours, finances and transport, provision of care and healthcare engagement. Most influencers acted as both barriers and facilitators, with the exception of finances and transport that only acted as a barrier, and healthcare engagement that acted as a facilitator.

**Conclusion** Understanding health influencers and care-seeking behaviours is recognised to facilitate appropriate community health programmes. Key influencers and care-seeking behaviours have been identified from rural Cambodia adding to the current literature. Where facilitators have already been established, they should be used as building blocks for continued change.

## Strengths and limitations of this study

► Establishing a professional research relationship with community health workers (CHWs) over time enabled CHWs to discuss sensitive topics and express their true opinions in focus group discussions (strength).

► Working with CHWs encouraged engagement in neonatal health at the community level (strength).

► The large groups of CHWs used for focus group discussions allowed them to build on their current relationships and trust, essential for ongoing healthcare engagement (strength).

► Focus group discussions were only conducted with one set of stakeholders, as it was assumed that CHWs would represent their community views. However, working with CHWs alone could have introduced bias (limitation).

► Lack of audio recording was an obstacle to data capture (limitation).

## INTRODUCTION

Neonatal mortality poses a significant global burden with 2.5 million neonatal deaths in 2017, one million of which are in the first day of life and 75% within the first week.[1] Although the Millennium Development Goals have helped drive a large reduction in under-5 child mortality over the past two decades, improvements in neonatal mortality have been less impressive, with neonatal mortality now accounting for 47% of under-5 child mortality globally.[1 2] Over 90% of neonatal deaths globally are in low/middle-income countries (LMICs), with the majority attributable to premature birth, intrapartum complications and infection.[1 3]

The Sustainable Development Goals have highlighted reduction of neonatal mortality as a priority and Sustainable Development Goal 3.2 aims to reduce the neonatal mortality rate below 12/1000 by 2030. Cambodia was successful in meeting Millennium Development Goals for reduction of under-5 child mortality and improving maternal health by investing in health systems, but still has some way to go to meet Sustainable Development Goal 3.2.[4] From the 2014 Cambodian Demographic Health Survey, the recorded neonatal mortality rate for Cambodia is 18/1000, however, rural areas have a higher mortality with the neonatal mortality rate in Preah Vihear province reported at 25/1000.[5]

Preventing neonatal deaths requires intervention at many levels. Community-based neonatal interventions have been shown to improve both early and late neonatal mortalities, stillbirth rate and health-seeking behaviours for unwell neonates.[6] Community interventions should be socially and culturally acceptable, practical, appropriate and,

in keeping with the communities' needs, working with community health workers (CHWs) allows these goals to be met. Participatory action research, alongside health system strengthening, offers a structure to facilitate interventions with all of these characteristics and has been used in numerous low resource settings since its conception in Bolivia with the Warmi project.[7] The WHO has recommended community mobilisation through facilitated participatory learning and action cycles for improvement of maternal and newborn health.[8]

This research was carried out within the Saving Babies Lives (SBL) programme being implemented by Angkor Hospital for Children, Cambodia, in partnership with the Kingdom of Cambodia Ministry of Health. Using participatory action research methodology, focus group discussions (FGDs) with CHWs aimed to identify influencers of neonatal health and care-seeking behaviours. Subsequent work will be carried out with the CHWs to explore and establish solutions to the identified problems, to action these solutions and to evaluate the outcome. In this manuscript findings of the first participatory action research phase, understanding the current situation in the province is presented.

## METHODS
### Study setting
Preah Vihear province in Northern Cambodia is one of the poorest and most rural in the country with an approximate population of 230 000 across 289 villages.[9] In 2016, there were 5292 health facility deliveries recorded, with an estimated 20% of births occurring at home (personal communication with Provincial Health Department, Preah Vihear).

The population is served by 28 health centres (HCs) and one referral hospital (RH). If a neonate requires high dependency or intensive care, they may be transferred to a tertiary care centre in the neighbouring province of Siem Reap.

The median distance of a HC from the RH is 92 km and serves a mean population of 8300 people. These figures are highly variable, with the largest HC serving 30 000 people spread across 41 villages, and the smallest serving a population of 393 from a single village. Villages also vary greatly in population size and distance from their HC. Many villages are more than an hour travel time away from their HC, with travel time increasing over the rainy season.

### Study design, population and data collection
The four HCs chosen to be included in this study, defined as clusters, were selected by the SBL team in collaboration with the provincial health department, with the intention that they should be representative of the whole province. The demographics of the chosen clusters are described in table 1.

In Cambodia, CHWs are referred to as Village Health Support Group workers, and each village is served by two Village Health Support Group workers. These volunteers are selected by village chiefs and commune chiefs, offering a network between the villagers, chiefs, communes and HC staff. For ease of understanding, Village Health Support Group workers will be referred by the generic term CHWs in this manuscript. The CHWs were the study population.

The SBL staff were trained in FGD methods in training days, which included theory, role play, individual sessions to develop facilitator skills and observed FGDs in the field. Continuous feedback was provided. Topic guides were created with the local team in English and, after ensuring understanding, translated into Khmer. FGDs were performed in Khmer with CHWs within monthly meetings already occurring at the HC. HC workers were not present in the meetings and they were conducted in private away from HC staff. Due to the large size of some CHW groups within clusters, groups were split, so that no more than 20 CHWs attended one FGD. Following consultation with the provincial health director and senior HC staff, it was felt that audio recording of FGDs was culturally unacceptable. Instead, to allow for adequate data capture, hand-written notes were made during FGDs, with the SBL team meeting immediately after to hold debrief sessions which were audio recorded. During these debrief sessions, all hand-written notes were translated into English, along with quotes from the FGD that were thought to be pertinent. Transcripts of all recorded debrief sessions were subsequently made.

| Table 1 | Demographics and attendance by cluster | | | | | | |
|---|---|---|---|---|---|---|---|
| Cluster | Population served by HC | Number of villages | Distance in km from HC to RH | Mean travel time from village to HC in minutes (dry and wet seasons) | Number of CHWs attending FGDs (female:male) | Number of FGDs conducted per month | Number of FGDs conducted in total |
| 1 | 7628 | 7 | 60 | 33:48 | 10 (8:2) | 1 | 4 |
| 2 | 21 142 | 33 | 92 | 40:44 | 48 (33:15) | 3* | 12 |
| 3 | 6358 | 9 | 70 | 20:35 | 16 (12:4) | 1 | 4 |
| 4 | 10 379 | 7 | 90 | 25:32 | 11 (9:2) | 1 | 4 |

*Given large number of CHWs in this cluster, three FGDs were conducted each month.
CHWs, community health workers; FGDs, focus group discussions; HC, health centre; RH, referral hospital.

Formal written consent for participation in FGDs was not taken from CHWs, as the provincial health department regarded CHW involvement as part of their routine work. Nevertheless, the voluntary and confidential nature of these meetings was emphasised and all CHWs agreed to participate. The CHWs were reimbursed for their travel costs to a value determined by the provincial health department.

### Researcher characteristics

All research staff performing FGDs were Cambodian paediatric nurses with clinical and research experience in Cambodia. These staff had developed a professional research relationship with CHWs over the previous 6 months during their monthly meetings. The research clinician involved in training, developing topic guides and data analysis was a British paediatrician with previous experience in LMICs and some qualitative research experience, supervised by the principal investigator who had extensive qualitative experience. All debriefs, topic guide iterations and initial analysis were performed together as a team.

### Data analysis

Data analysis was conducted iteratively and topic guides for subsequent meetings were developed to capture emerging themes. All translated notes were imported into the R-based Qualitative Data Analysis package.[10 11] Thematic content analysis was used and themes were derived from the data. An inductive approach was taken to the analysis with no prior assumptions made about the data. The study team discussed and agreed on emerging themes.

### Patient and public involvement

Given the nature of the study, the public were involved throughout. Public engagement is at the core of participatory action research, facilitating communities to find problems and address them themselves. Establishing communities' perception of neonatal health and their care-seeking behaviours is essential to design and implement successful future interventions in neonatal health.

### RESULTS

A total of 24 FGDs were conducted with 85 CHWs, 62 women and 23 men, over 4 months which allowed the development and exploration of the following themes:

► Experience and impact of neonatal health issues.
► Parental healthcare-seeking behaviours for neonates.
► Perceived improvements in neonatal health over time and facilitators of these improvements.
► Persisting barriers to good neonatal health.

### Experience and impact of neonatal health issues

All CHWs could recall experiences of stillbirth, premature deliveries, neonatal illness and death.

at seven months gestation, one mother was working hard with heavy manual labour, she stopped feeling her baby moving and went to the RH, she was told the baby had died in the womb and went on to have induction of labour and the baby was born dead. (Cluster 2, CHW)

[One CHW] had a premature baby, she was very small and born at 6 months 17 days, it was difficult to take care of her and she was often getting sick, but she is now working as a teacher and has her own child. (Cluster 2, CHW—female)

The impact of neonatal health problems was reported to be felt at a family and community level. All groups reported social and financial burdens on families affected by poor neonatal health and the negative impact on the village's economic status. Some groups reported community-level criticism of villages with poor neonatal health.

CHWs also recognised blame from elders for neonatal illness, death and stillbirth.

If mother has a stillbirth, elders will blame the mother for not taking care of herself in pregnancy or working too hard. If a baby is sick and the mother doesn't recognise this, there will be blame from the grandmother. (Cluster 2, CHW)

### Care-seeking behaviours for the neonate

Decisions for seeking care were stated to lie predominantly with the parents and, in particular, the mother, but it was found that she would often seek advice from CHWs or family members. The choice of care sought varied widely depending on multiple factors: cost of user fees and transport; travel distance; perceived severity of illness and level of trust in local facilities. The primary caregiver would usually be accompanied to a health facility by another family member or a CHW.

Mothers don't apply traditional beliefs to the newborn baby – as parents know a lot now and the baby is very small, so they go to CHWs house and discuss, CHWs will then advise to go to HC or RH and CHWs will go with them. (Cluster 1, CHW)

Discussions on the use of traditional practices differed between clusters and even between villages within clusters. Decisions made in choosing between modern medicine and traditional practices were stated to have changed over the years, with most CHWs reporting that the first choice of care for neonates would be modern medicine. However, if improvement was not seen quickly, villagers would often go on to seek traditional practices. Cambodian traditional medicine healers are referred to locally as Kru Khmer.

Some people will go to health facility and Kru Khmer – nowadays they will go to health care worker first and then if no quick improvement they will go to Kru Khmer. (Cluster 4, CHW)

Most CHW's demonstrated an understanding of the potential negative impact of traditional practices. For example, most groups associated poor umbilical care with tetanus. However, some traditional practices, such as tiger balm application, were felt to have positive impact and are still used.

### Perceived improvements in neonatal health and facilitators

All groups felt strongly there had been an improvement in neonatal health in their villages and that fewer problems were now seen. Facilitators to this change were identified as knowledge, sociocultural behavioural change, provision of care and healthcare engagement.

#### Knowledge

CHWs' knowledge of neonatal health was better than was expected by the SBL team. Knowledge had been acquired over the years from different non-governmental organisation (NGO) teaching initiatives and HC outreach programmes. They valued their knowledge and took pride in teaching villagers. All CHWs reported a sense of responsibility in sharing their knowledge with villagers and saw the impact of improved health in their community.

> CHWs feel a responsibility to visit mothers in pregnancy and after the baby is born to provide education. (Cluster 2, CHW)

#### Sociocultural behaviours

All CHWs reported changing attitudes to traditional practices for neonates, due to education from CHWs and HC staff and improvement in the general population education level. As already stated, most villagers would choose modern medicine over traditional practices for neonates.

> [CHWs] feel parents now pay more attention to babies' health and if they notice anything unusual they take them to the HC, in the past they might have taken them to traditional healer first, as a result they feel there is no longer a problem with babies dying in the villages. (Cluster 3, CHW)

#### Provision of care

The increased number of HCs and better access to antenatal care were felt to have been pivotal in improving community health.

> in the past when they had no HC, they were concerned about safe delivery for the first pregnancy, and they went to traditional birth attendant. (Cluster 4, CHW)

#### Healthcare engagement

All CHWs reported the importance of their educational and supportive role. They also felt women gained trust in them as they visited more often and were proud of the improvements they had facilitated. Local authorities, such as commune councils and women's affairs groups, were also felt to play an important role in health

improvements, and all clusters mentioned current and past NGOs' involvement in positively impacting health at a community and health facility level.

> [CHWs] provide information and educate villagers, [CHW] have taken a role in changing healthcare. (Cluster 3, CHWs)

> a story of a difficult delivery, baby delivered after 4 pulls by vacuum…the baby was blue after birth and needed bagging for 10 min, then the baby cried… baby survived…Midwife says they had just had resuscitation training from SBL so they did what they had been taught and the baby survived. (Cluster 3, CHW)

### Persisting barriers to neonatal health

Despite the improvements described above, all clusters felt barriers to neonatal health persisted. With each group independently highlighting the same barriers, barriers identified were knowledge, sociocultural behaviours, finances and transport and provision of care.

The CHWs felt the responsibility for addressing these barriers lay with many stakeholders: at the village level (family, villagers, village chief); at the health facilities (both HC and RH); with authorities (commune chief, women's affairs groups, police) and with the NGOs.

#### Knowledge

Opinions on recognition of danger signs in the neonate or labouring mother differed slightly between clusters. Most CHWs felt that recognition of danger signs was good among experienced mothers but that new mothers would not recognise all danger signs. All CHWs saw it as their responsibility to teach mothers' danger signs.

> Some mothers can't recognise danger signs like drowsiness – the mother just thinks that baby is quiet and easy to look after. (Cluster 3, CHW)

Some CHWs also identified gaps in their own knowledge such as caring for the premature baby, effects of traditional practices, danger signs in the labouring mother and teaching skills.

#### Sociocultural behaviours

Sociocultural behaviours impact significantly on the decision to seek care, with choice of care still a concern in some villages. Although rare, some villages still choose traditional practices, particularly if parents do not see a fast response to modern medicine. Villagers' decisions to seek care were also influenced by their previous experiences of HCs, including quality of care, staff attitudes and blame from HC staff.

> There is sometimes blame from the HC staff if the patient comes too late to the HC, parents might have a sick baby at home for a few days and then when they get worse they are scared to go to the HC because of blame from staff for bringing the baby too late. (Cluster 4, CHW)

## Financial and transport barriers

Financial and transport barriers were felt to be significant in all villages. Cost of transport to health facility, user fees at HC and RH, expenses while away from home, cost of referral, loss of potential earnings and resultant debt were all reported as financial concerns. Transport concerns were reported as the distance from the health facility, availability of transport, road conditions, time of day and seasonality. All these factors were reported to impact on villagers' decisions to seek care, choice of transport and choice of care.

> Parents do not want to waste money on minor illness, but when they become seriously unwell or in an emergency then they will spend money. (Cluster 1, CHW)

> The road is ok in the dry season but during the rainy season it is not possible to access some villages. (Cluster 3, CHW)

## Provision of care

Acceptability of health facilities differed between clusters depending on which HC provided their care. Concerns raised surrounded HC equipment and supplies, staffing hours and staff attitudes. These concerns were reported to impact villagers' decisions to seek care.

> HC treatment is limited – lack of equipment, medication shortage, waiting for long time for review as the HC staff are at lunch. (Cluster 3, CHW)

> HC staff treat people differently between poor and rich families, they give less compassionate care. (Cluster 2, CHW female)

## DISCUSSION

From FGDs with CHWs, and analysis of the data, we have gained an understanding of factors that influence neonatal health and healthcare-seeking behaviours in Preah Vihear province. Key influencers were identified as knowledge, sociocultural behaviours, finances and transport, provision of care and healthcare engagement. Some of these influencers have acted as facilitators while others remain barriers. The barriers identified are in keeping with Thaddeus and Maine's three-delay model: (1) delay in decision to seek care, (2) delay in reaching healthcare and (3) delay in receiving adequate care[12] (see figure 1). Although the three-delay model was originally created for exploring maternal death, it has recently been used successfully to address delays in perinatal and neonatal deaths and to provide a useful framework to address barriers in neonatal health.[13–15]

With half of the global neonatal deaths happening at home and poor accessibility of health facilities in rural areas, good community knowledge and health engagement are key for early recognition of the sick neonate.[16] The Cambodian 'Community Participation Policy for Health' sees CHWs as an essential link between HC and community and key for health promotion.[17] In Cambodia,

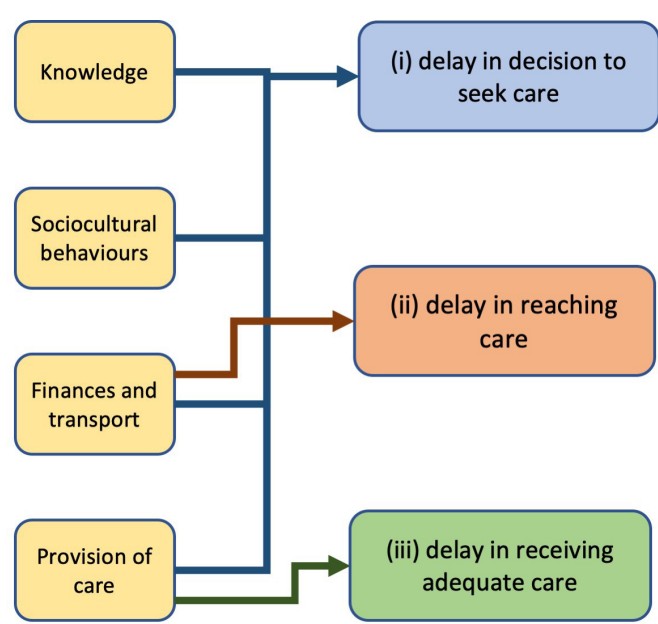

**Figure 1** Barriers to seeking care identified alongside the Thaddeus and Maine's three-delay model: (i) delay in decision to seek care, (ii) delay in reaching healthcare and (iii) delay in receiving adequate care.

CHWs have been shown to improve health engagement, however, their use is often reliant on NGO support.[18] We found CHWs knowledge of neonatal health to be good. They are engaged with neonatal health, taking on responsibility for educating mothers. In addition, they see sociocultural behavioural change as an important facilitator to improving neonatal health. With the right support, CHWs are ideally placed to encourage health engagement and further community education, however, NGO support is not always sustainable. Ozano et al highlight challenges for CHWs in Cambodia in keeping with global CHW studies: inadequate financial remuneration, training and supervision.[18 19] Government ownership has been reported as essential for sustainable success of CHW programmes, further highlighting the importance of ongoing governmental recognition and support for CHW in Cambodia.[20 21]

While CHW neonatal health knowledge was better than expected, they themselves identified missing knowledge and requested teaching. Confidence in knowledge application and teaching also needs to be improved. Their suggestions will lead to a targeted teaching programme which they will give to villagers.

Reported financial and transport barriers are in keeping with many LMIC settings. Although Cambodia's national IDPoor programme covers healthcare-user fees for those issued with an equity card, it has only recently addressed other barriers related to finance, such as transport costs and loss of earnings.[22] In June 2019, a new scheme was introduced under the IDPoor programme, which provides financial incentive for mothers to attend HCs for antenatal, delivery and postnatal care. This could act as a facilitator for the poorest,

but barriers will remain for those without an equity card. A systematic review of community-based loan funds for obstetric emergencies suggests they can have positive effects, and case studies in rural Cambodia have used targeted health vouchers and funds to improve access to skilled birth attendants with success.[23 24] These may present options in this setting. As with many rural settings, difficulties with transport and access to health facilities are marked. Distance, seasonal road conditions and availability of appropriate transport all act as barriers. Developing transport infrastructure and appropriate emergency transport options is essential.

Many financial barriers are challenging to address, while others are potentially simpler. We identified maternal misconceptions around user fees at health facilities as a cause of anxiety when deciding to seek care. This observation has been made elsewhere in Cambodia.[25] Given standardised Ministry of Health costs for health facility care, this could easily be addressed by improved communication between HC staff, CHWs and villagers.

A national health coverage plan was introduced in Cambodia in 1996 with the government developing and rebuilding health facilities throughout the country. CHWs saw the introduction of HCs as fundamental to the improvement of neonatal health but, when explored further, provision of care is still a barrier. Equipment and supplies at HCs were often perceived to be poor. Some of these perceptions were justifiable, for example, essential medications being out of stock. On the other hand, some concerns were less well described or objective and might not have correlated with quality of service provision. The poor attitude of staff at some health facilities is reported to impact on villagers' decisions to seek care and can influence trust in health facilities. Similar concerns have been raised by Matsuoka *et al* in Cambodia.[25] Compounded with a documented mistrust of outsiders and authorities in Cambodia, in the context of geopolitical events over the past half century, this will be difficult to address.[26] CHWs report villagers' trust in them is high and, as they act as an essential link between the villagers and HCs, building on this may allow for improvements in trust between villagers and the HC. Staff communications, blame from HC staff and discrimination need to be addressed at an institutional level.

Commune councils, local authorities and NGOs were identified as key enablers to addressing barriers in neonatal health and their continued engagement is essential to ongoing success. Reliance on NGOs is not uncommon in LMICs but, to ensure sustainable success, changes must be implemented alongside the Ministry of Health and the community.

Understanding these barriers allows them to be addressed appropriately at the community level. CHWs will continue to meet, facilitated by SBL, and use the participatory action research structure to explore and implement solutions and evaluate their impact.

## Study limitations

FGDs were only conducted with one set of stakeholders as it was assumed that CHWs would represent their community views, however, working with CHWs alone could have introduced bias. Although the lack of audio recording was an obstacle to accurate data capture, it did enable CHWs to feel more comfortable discussing sensitive topics and expressing their true opinions. The large group size, atypical for FGDs, was a potential limitation. However, this allowed for groups of CHWs to build on their current relationships and trust, essential for ongoing healthcare engagement.

## CONCLUSION

Understanding health influencers and care-seeking behaviours is recognised to facilitate appropriate community health programmes. Key influencers and care-seeking behaviours have been identified from rural Cambodia adding to the current literature. Some key influencers act as both facilitators and barriers. Where facilitators have already been established, they should be used as building blocks for continued change.

**Acknowledgements** The authors would like to extend their thanks to all the Village Health Support Group workers in Preah Vihear province for their contributions to focus group discussions, the Ministry of Health, Cambodia and the Provincial Health Department of Preah Vihear Province for their support of the study.

**Contributors** GW, CT and MP conceived the study. GW, KP, DL, DV and SK were responsible for data collection. GW did the data analysis and prepared the first draft of the manuscript. All authors reviewed and contributed to manuscript revisions. All authors read and approved the final manuscript.

**Funding** Saving Babies Lives Community projects during this time were funded by Angkor Hospital for Children, Manon, Civil Society in Development, Vitol Foundation and the Wellcome Trust— Wellcome Public engagement fund.

**Competing interests** None declared.

**Patient and public involvement** Patients and/or the public were involved in the design, or conduct, or reporting, or dissemination plans of this research. Refer to the Methods section for further details.

**Patient consent for publication** Not required.

**Ethics approval** The study was approved by the Cambodian National Ethics Committee (229 NECHR) and Oxford Tropical Research Ethics Committee (OxTREC, 547-17).

**Provenance and peer review** Not commissioned; externally peer reviewed.

**Data availability statement** Data are available upon reasonable request.

**ORCID iD**
Gabriella Watson http://orcid.org/0000-0001-5873-534X

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
