## [Reviewer comments · BMJ Open]

ARTICLE DETAILS

TITLE (PROVISIONAL)	Barriers and facilitators to neonatal health and care seeking behaviours in rural Cambodia, a Qualitative study
AUTHORS	Watson, Gabriella; Patel, Kaajal; Leng, Daly; Vanna, Dary; Khut, Sophanou; Prak, Manila; Turner, Claudia

VERSION 1 - REVIEW

REVIEWER	David K Wyant The Jack C. Massey College of Business Belmont University, Nashville Tennessee, USA
REVIEW RETURNED	14-Dec-2019

GENERAL COMMENTS	This paper would provide a valuable contribution to the literature as it is currently written. However, either in this paper or perhaps a subsequent paper there are several subjects that if addressed could add value. The abstract could be strengthened by indicating the year that the focus groups were constructed. This would be helpful to the extent that the situation is changing over time. It also might be good to have "Cambodia" as a key word. The authors have identified a number of barriers and noted the "barriers are in keeping with Thaddeus and Maine's three -delay model". The authors might show how these barriers would be grouped within the model, perhaps in a diagram. This might show relationships between the barriers, including the extent to which these barriers reinforce each other. A model can also make it easier to follow the text by providing structure. Discussion of barriers could be grouped by whether they delay decisions, delay reaching care, or delay receiving care. Another model that comes to mind is Ron Anderson's behavioral health model, which might be an alternative. An overall model would be helpful, if it fits the situation. Probably this is left for another study, but to what extent might an information technology solution be potentially helpful? This might be aimed at community health workers or at women who may soon be mothers. I don't know the situation in Cambodia, but in one Guatemala town I visited, although smart phone ownership is relatively low for people with low incomes, many low income people have access through relatives or internet cafes. If there was a targeted program available for free on the internet what share of the population you saw might be able to access it one way or another? Alternatively, a program that provided smart phones to CHWs might not be very expensive. I recognize that this was not your study
---

	question, but my question for this project is did you get any impression of how an IT based program aimed at CHW's or potential mothers might address any of the barriers you identified?
--	---

REVIEWER	Calistus Wilunda African Population and Health Research Center
REVIEW RETURNED	03-Feb-2020

GENERAL COMMENTS	Thank you for the opportunity to review this manuscript on influencers of neonatal health and healthcare seeking behaviour in Cambodia. The paper addresses an important topic and is generally well written. However, I have the following major concerns:  1. In the Conclusions, the authors make a sweeping statement that is not supported by the data from this study. There is no evidence from this study that "Understanding health influencers and care seeking behaviours in a community is fundamental to establishing an effective and sustainable community health programme". Making such a conclusion is beyond the scope of this study. 2. Collecting data from only CHWs is a major limitation of this study. There was no triangulation of data from other sources, which casts doubts on the conclusions. Although this has been acknowledged, I think it is insufficient to just acknowledge such a limitation and move on. Triangulation of findings from different groups of respondents is key in ensuring the validity of findings from such a qualitative study. 3. There are some sub-themes without any supporting quotes from respondents. 4. Data collection by Cambodian paediatric nurses with a professional relationship with CHWs might have biased the study findings. It is likely that CHWs did not express themselves freely especially on matters relating to nurses and the health system. Although issues on "Provision of care" emerged, it seems like this was an "under-response". Conducting FGD sessions in HCs might have also limited freedom of expression by CHWs resulting into skewed responses. Minor comments  5. Page 6: Please state explicitly that the study population is CHWs 6. Page 6: Please clarify the method was used to select clusters. It seems like the method used (though not stated) does not match the intention (to ensure representativeness). 7. In Table 1, include a column with the total number of FGDs conducted 8. Page 6, line 52: The following statement is not clear: "The SBL staff were trained in PAR and FGD methods in training days..." 9. Page 7, line 24: This statement does not make sense. Did the ethics committee waive the need for written informed consent? 10. Page 7, line 43-45: In addition to experience in LMICs, please clarify whether the British Paediatrician had prior experience in qualitative research. 11. In the results section, include data on the sociodemographic characteristics of the respondents. 12. Page 13, lines 16-18. Please use the past tense. 13. Page 17, line 23-25: this is important information but it is not relevant to the present study as it is not linked to any of the objectives 14. Although the manuscript is well written, it can benefit further from light English editing. Here are some of the errors identified:
---

	Page 4, line 36 and elsewhere in the manuscript. Insert a comma after "however" Page 5, line 38: insert a comma after "2016" Page 6, line 42: use "the village chief" or village chiefs in this statement "These volunteers are selected by village chief and... " Page 15, lines 17-19: check the punctuation of this statement. Page 15, line 51: insert a comma after 2019
--	--

VERSION 1 – AUTHOR RESPONSE

Reviewer 1 comments and responses:

Comments to the Author:

This paper would provide a valuable contribution to the literature as it is currently written. However, either in this paper or perhaps a subsequent paper there are several subjects that if addressed could add value.

Comment: The abstract could be strengthened by indicating the year that the focus groups were constructed. This would be helpful to the extent that the situation is changing over time. It also might be good to have "Cambodia" as a key word.

Response: We have added the year the focus groups were constructed into the abstract and we have added Cambodia as a key word.

Comment: The authors have identified a number of barriers and noted the "barriers are in keeping with Thaddeus and Maine's three -delay model". The authors might show how these barriers would be grouped within the model, perhaps in a diagram. This might show relationships between the barriers, including the extent to which these barriers reinforce each other. A model can also make it easier to follow the text by providing structure. Discussion of barriers could be grouped by whether they delay decisions, delay reaching care, or delay receiving care. Another model that comes to mind is Ron Anderson's behavioural health model, which might be an alternative. An overall model would be helpful, if it fits the situation.

Response: We have developed a diagram to illustrate the barriers and their relationships and to support the text structure. We thank the reviewer for suggesting the Andersen healthcare utilisation model, a conceptual model to understand why families use health services. It was designed in the 1960s consisting of 'predisposing characteristics, enabling resources, need, use of health services', and further reviewed in 1995. We feel the three-delay model is more representative of our data.

Comment: Probably this is left for another study, but to what extent might an information technology solution be potentially helpful? This might be aimed at community health workers or at women who may soon be mothers. I don't know the situation in Cambodia, but in one Guatemala town I visited, although smart phone ownership is relatively low for people with low incomes, many low income people have access through relatives or internet cafes. If there was a targeted program available for free on the internet what share of the population you saw might be able to access it one way or another? Alternatively, a program that provided smart phones to CHWs might not be very expensive. I recognize that this was not your study question, but my question for this project is did you get any impression of how an IT based program aimed at CHW's or potential mothers might address any of the barriers you identified?

Response: This study was the situational analysis in-bedded in a much larger study, Saving Babies Lives, which uses participatory action research with CHWs, as such it is anticipated that one solution proposed and implemented may well be smart phones or mobile devices. However currently it is

anticipated that the use of smart phones in our study area, is not sufficiently widespread to provide adequate cover. We did not focus on information technology specifically in FGDs and it did not come up as a theme, as such it was not discussed in the results or analysis. However, we appreciate the reviewers comments and the potential role for m-health going forward.

Reviewer 2 comments and responses:

Comments to the author:

Thank you for the opportunity to review this manuscript on influencers of neonatal health and healthcare seeking behaviour in Cambodia. The paper addresses an important topic and is generally well written. However, I have the following major concerns:

Major comments:

1. In the Conclusions, the authors make a sweeping statement that is not supported by the data from this study. There is no evidence from this study that "Understanding health influencers and care seeking behaviours in a community is fundamental to establishing an effective and sustainable community health programme". Making such a conclusion is beyond the scope of this study.

Response: We regret that our phrasing in the conclusion has led to a misunderstanding. To be clear, we did not mean to infer our data supported this statement. It is recognised that understanding health influencers and care seeking behaviours are important in policy and programme planning. As such identifying influencers and care-seeking behaviours in rural Cambodia adds to the literature and could support future community health programme development. We have amended the conclusion to make this clear.

2. Collecting data from only CHWs is a major limitation of this study. There was no triangulation of data from other sources, which casts doubts on the conclusions. Although this has been acknowledged, I think it is insufficient to just acknowledge such a limitation and move on. Triangulation of findings from different groups of respondents is key in ensuring the validity of findings from such a qualitative study.

Response: We agree with the reviewer that lack of triangulation is a limitation of the study, and the authors feel that this has been stated clearly in the manuscript.

3. There are some sub-themes without any supporting quotes from respondents.

Response: There was no supporting quote for the sub-theme 'Care seeking behaviours', we have amended this and attached supporting quotes from the respondents.

4. Data collection by Cambodian paediatric nurses with a professional relationship with CHWs might have biased the study findings. It is likely that CHWs did not express themselves freely especially on matters relating to nurses and the health system. Although issues on "Provision of care" emerged, it seems like this was an "under-response". Conducting FGD sessions in HCs might have also limited freedom of expression by CHWs resulting into skewed responses.

Response: We agree with the reviewer the above could bias the findings, however, our use of the phrase 'professional relationship' could have been misleading; all Cambodian Paediatric nurses, were members of the research team and nurses at Angkor Hospital for Children, a Non-Governmental-Organisation hospital in Siem Reap, as such they did not have professional relationship with the CHW other than within the research programme. The research nurses were an independent party from the

Health centres. Focus group discussions were carried out at the Health Centers, on request of the Provincial health department to co-incide with CHW monthly responsibilities. Health Center staff were not allowed in the focus group discussions and they were wither conducted in closed meeting rooms within the Health Center, or outside the Health Center under the shade of trees away from the Health Center staff. We have amended the manuscript to clarify both these points.

Minor comments:

5. Page 6: Please state explicitly that the study population is CHWs

Response: we have amended the study design to explain the study population were CHWs.

6. Page 6: Please clarify the method was used to select clusters. It seems like the method used (though not stated) does not match the intention (to ensure representativeness).

Response: As outlined in the study design clusters were selected by the provincial health department to ensure they would be representative of the whole province.

7. In Table 1, include a column with the total number of FGDs conducted

Response: we have inserted a further column with the total number of FGDs conducted

8. Page 6, line 52: The following statement is not clear: "The SBL staff were trained in PAR and FGD methods in training days..."

Response

Apologies PAR was an error, we have amended it accordingly. SBL staff were trained in how to conduct focus group discussions, including the use of role play and focused sessions with individuals to develop their facilitator skills, and focus group discussions were observed in the field with weekly feedback given to facilitators.

9. Page 7, line 24: This statement does not make sense. Did the ethics committee waive the need for written informed consent?

Response: For the reasons stated written informed consent was not taken, which was agreed by both the Cambodian National Ethics committee and the Oxford Tropical Research Ethics committee. Consent was inferred when CHW attended the meetings and it was made clear all participation in the FGD was confidential and voluntary. This is outlined in the study design.

10. Page 7, line 43-45: In addition to experience in LMICs, please clarify whether the British Paediatrician had prior experience in qualitative research.

Response: We have amended this to explain the British Paediatrician had some qualitative research experience and was supervised by the study's principal investigator who had extensive experience.

11. In the results section, include data on the sociodemographic characteristics of the respondents.

Response: We had previously included the demographics of the respondents in table 1, and have now amended the results section to also include general demographics. CHW were anonymised at point of creating transcripts, as such we have no individual demographics for the respondents.

12. Page 13, lines 16-18. Please use the past tense.

Response: We have amended this to the past tense.

13. Page 17, line 23-25: this is important information but it is not relevant to the present study as it is not linked to any of the objectives

Response: We appreciate the reviewer's comment and have removed this.

14. Although the manuscript is well written, it can benefit further from light English editing.

Here are some of the errors identified:

Page 4, line 36 and elsewhere in the manuscript. Insert a comma after "however"

Page 5, line 38: insert a comma after "2016"

Page 6, line 42: use "the village chief" or village chiefs in this statement "These volunteers are selected by village chief and... "

Page 15, lines 17-19: check the punctuation of this statement.

Page 15, line 51: insert a comma after 2019

Response: The above grammar has been corrected in line with the reviewer's comments.

VERSION 2 – REVIEW

REVIEWER	David K Wyant Belmont University Nashville Tennessee, USA
REVIEW RETURNED	02-Apr-2020

GENERAL COMMENTS	First, I want to pass on my regret for being overdue on this. My city and University went on lockdown and we needed to transfer all classes to online. I had to stop other activities to focus on that part of my job. I know I committed to an earlier date but the delay was unavoidable. 1. I made suggestions for an earlier version of this manuscript. The present draft has taken those comments into account. As such, I have only minor suggestions at this point. 2. There are too many abbreviations. It was hard for me to keep track. Often in the text, the abbreviation is provided and then only used 1 or 2 times for the rest of the paper. For example, VHSGs. Why not just write these out instead of abbreviations? I'm not certain that AHC is used at all after it introduced. Also please write out MOH. It may not be understood out of context. When an abbreviation is provided, the implication is that a term is going to be used frequently, so space will be saved by using the abbreviation. The reader may think this term is somewhat of a "main character" in the discussion that will follow. But there are too many main characters. There may be some cases, such as NGO, where the reader is more likely to have encountered the abbreviation, so fine to use it. Do you believe that is the case for MDG, SDG, NMR and LMIC? If so fine, otherwise just write them out the few times they are mentioned. Perhaps you want PAR and SBL to be known by their initials. If so OK, use the abbreviation, otherwise please write them
---

	out. Others are used frequently so they are ok, for example CHW, HC, RH, FGD. 3. There are misspellings on page 8 of 46 “Haelth” and on page 15 of 46 “mone” . 4. There is a reference to Kru Khmer in a quote on my page 11 of 46. A sentence before the quote could explain Kru Khmer. 5. In several places minor changes could make the text flow more smoothly a. In the box that shows strength and limitations, for the first point it is not clear who the relationship and discussion involve (who is “them”?). Similarly for the third point , what is the group type that is “large size” b. In the introduction, the two times that there is a backward referring “this” in the following are unclear, because there are so many phrases strung together. What is “this”: Community interventions should be socially and culturally acceptable, practical, appropriate and, in keeping with the communities’ needs, working with community health worker’s allows this. Participatory Action Research (PAR), alongside health system strengthening, offers a structure to facilitate this... Is “this” something like “interventions with all of these characteristics” c. The last sentence before methods needs editing :”In this manuscript finding in the first PAR phase, understanding the current situation in the province are presented.” d. On my page 7 of 46 In the sentence beginning “The SBL staff” there is an extra “and” at “and role play”. e. The last sentence of the conclusion has an extra “to”. Thank you for sharing this manuscript with BMJ Open.
--	--

REVIEWER	Calistus Wilunda African Population and Health Research Center, Kenya
REVIEW RETURNED	07-Mar-2020

GENERAL COMMENTS	The authors have adequately addressed my comments. However, if accepted, the authors should read their proof carefully and correct any typos or missing punctuation marks. For example: Page 2, line 45: There is a typo here “..they should to be used as building blocks” Page 5, line 24: It seems like a comma is missing after “province”
---

VERSION 2 – AUTHOR RESPONSE

Reviewer 1 comments and responses:

Comments to the Author:

I made suggestions for an earlier version of this manuscript. The present draft has taken those comments into account. As such, I have only minor suggestions at this point.

Comment:

There are too many abbreviations. It was hard for me to keep track. Often in the text, the abbreviation is provided and then only used 1 or 2 times for the rest of the paper. For example, VHSGs. Why not

just write these out instead of abbreviations? I'm not certain that AHC is used at all after it introduced. Also please write out MOH. It may not be understood out of context. When an abbreviation is provided, the implication is that a term is going to be used frequently, so space will be saved by using the abbreviation. The reader may think this term is somewhat of a "main character" in the discussion that will follow. But there are too many main characters. There may be some cases, such as NGO, where the reader is more likely to have encountered the abbreviation, so fine to use it. Do you believe that is the case for MDG, SDG, NMR and LMIC? If so fine, otherwise just write them out the few times they are mentioned. Perhaps you want PAR and SBL to be known by their initials. If so OK, use the abbreviation, otherwise please write them out. Others are used frequently so they are ok, for example CHW, HC, RH, FGD.

Response:

Thank you for highlighting this, we agree and have removed all but essential and frequently used abbreviations.

Comment:

There are misspellings on page 8 of 46 "Haelth" and on page 15 of 46 "mone" .

Response:

These have been corrected.

Comment:

There is a reference to Kru Khmer in a quote on my page 11 of 46. A sentence before the quote could explain Kru Khmer.

Response:

We have added a sentence prior to the quote explaining the Kru Khmer is the local term for a Cambodian traditional medicine healer.

Comment:

In several places minor changes could make the text flow more smoothly

a. In the box that shows strength and limitations, for the first point it is not clear who the relationship and discussion involve (who is "them"?). Similarly for the third point , what is the group type that is "large size"

b. In the introduction, the two times that there is a backward referring "this" in the following are unclear, because there are so many phrases strung together. What is "this":

Community interventions should be socially and culturally acceptable, practical, appropriate and, in keeping with the communities' needs, working with community health worker's allows this.

Participatory Action Research (PAR), alongside health system strengthening, offers a structure to facilitate this...

Is "this" something like "interventions with all of these characteristics"

c. The last sentence before methods needs editing : "In this manuscript finding in the first PAR phase, understanding the current situation in the province are presented."

d. On my page 7 of 46 In the sentence beginning "The SBL staff" there is an extra "and" at "and role play".

e. The last sentence of the conclusion has an extra "to".

Response:

Thank you for these comments, we have addressed all of the above suggestions and amended the text accordingly.

Reviewer 2 comments and responses:

Comments to the author:

The authors have adequately addressed my comments. However, if accepted, the authors should read their proof carefully and correct any typos or missing punctuation marks. For example:

Page 2, line 45: There is a typo here ".they should to be used as building blocks"

Page 5, line 24: It seems like a comma is missing after "province"

Response:

We have corrected the examples and proofread the manuscript carefully for spelling and grammar which has been corrected accordingly.